# A Protocol to Assess the Welfare of Patagonian Huemul (*Hippocamelus bisulcus*) in Conservation Centers

**DOI:** 10.3390/ani13152495

**Published:** 2023-08-02

**Authors:** Enrique Bombal, Xavier Manteca, Oriol Tallo-Parra

**Affiliations:** 1Zoo Animal Welfare Education Centre (ZAWEC), Animal Welfare Education Centre (AWEC), Universitat Autònoma de Barcelona, 08193 Barcelona, Spain; enrique.bombal@gmail.com (E.B.); oriol.tallo@uab.cat (O.T.-P.); 2Department of Animal and Food Science, School of Veterinary Science, Universitat Autònoma de Barcelona, 08193 Barcelona, Spain

**Keywords:** behavior, captivity, *Hippocamelus bisulcus*, huemul, animal-based measures, welfare assessment

## Abstract

**Simple Summary:**

Animal welfare is an important aspect of conservation programs for endangered species. Wild species can be bred and kept in captivity but, unlike domestic animals, there is a lack of welfare-assessment protocols for most of these species. In this study, we developed a protocol for the assessment of the welfare of Patagonian huemul (*Hippocamelus bisulcus*) in conservation, centers. We gathered the existing research on the behavior, ecology, conservation and captive management for this species. We used a welfare-assessment protocol for cattle as our starting point and suggested 23 indicators to assess welfare in Patagonian huemuls. This proposed protocol, which is the first protocol for assessing Patagonian huemul welfare, is rigorous and systematic, but also simple and practical. Further research is needed to validate the protocol in conservation centers; nevertheless, this protocol could be used as a basis for the development of new welfare-assessment protocols for other deer species.

**Abstract:**

Animal-welfare-assessment protocols are important for identifying welfare problems in conservation programs. This study aimed to develop a baseline welfare protocol for the assessment of the welfare of Patagonian huemuls (*Hippocamelus bisulcus*) in conservation centers. This protocol is based on the Welfare Quality^®^ (WQ) framework for cattle and was developed with the consideration of the available research on the behaviors, ecology, conservation, and captive management of this species, as well as welfare-assessment protocols for other ungulate species. As a result, the protocol was specifically developed for Patagonian huemuls and included four principles, 12 criteria, and 23 animal- and resource-based indicators. The twelve criteria of the WQ protocol were reduced to nine, and three new criteria were added because they were both feasible and essential for welfare assessment in captive Patagonian huemuls. This protocol is mainly intended to identify welfare problems in endangered species in the context of conservation centers (reproduction, rescue, rehabilitation, or treatment centers). Thus, the aggregation of different measures to obtain a global score was not proposed. However, a scoring system that assigns a value on a 0–2 scale (0 = no welfare concern; 1 = welfare concern; 2 = urgent welfare concern) was proposed for each category. Although further research is still needed to fully validate the protocol, this is the first development of a protocol to assess Patagonian huemul welfare, and it can be used as a basis for the development of welfare-assessment protocols for other deer species in captivity.

## 1. Introduction

Animal welfare is a priority in ex situ conservation programs [1]. Guaranteeing the best welfare status of captive animals is important not only for ethical reasons, but also to ensure that captive breeding programs are robust and sustainable [2]. Animal welfare comprises the emotional state, physical health, and behavior of animals [3]. Welfare is an attribute of each individual and cannot be measured directly, but it must be evaluated using indicators, which are variables that can be measured objectively and provide information on the welfare of animals [1]. Welfare-assessment protocols combine several indicators and include a description of how to measure each indicator using simple surveys, enclosure inspections, and animal observations [3]. Welfare indicators can be divided into two main categories: animal-based indicators and environment-based indicators. Animal-based indicators include variables that are measured directly on animals (behavior, overall appearance, health, and physiological parameters) [1,3], whereas environment- or resource-based indicators assess the environment surrounding the animal but not the animal itself (e.g., water provision, the size and composition of the social group, and environmental enrichment). Welfare Quality^®^ (WQ) protocols (Lelystad, The Netherlands) are among the most commonly used for evaluating the welfare of farm animals (bovines, pigs, and poultry) [4], and they have been used as a starting point to develop protocols for zoo and companion animals. The indicators of the WQ protocols are grouped into four principles and twelve welfare criteria, and these principles coincide with the four measurable domains of the five-domain model (nutrition, physical health, comfort, and behavior) [1].

The *Hippocamelus bisulcus* (Patagonian huemul), an endemic species in Chile and Argentina, is part of the Artiodactyla order, the Ruminantia suborder, the Cervidae family, and the Capreolinae subfamily, and it is the southernmost deer in the world [5,6,7]. Currently, it inhabits fragmented areas in the southern regions of both Chile and Argentina, ranging from sea level to 3000 m above sea level, and areas covered mainly by Nothofagus forests, dwarf shrubs, rocky slopes, and meadows [6]. Its diet is strictly herbivorous, and although a preference for leaves, bush shoots, trees, and grasses has been described [6], this preference can vary substantially from one subpopulation to another [7].

This species has been declared endangered worldwide, according to the International Union for Conservation of Nature (IUCN)’s Red List of Threatened Species. The Patagonian huemul is considered Endangered B2ab (i, ii, iii, iv, v) C2a (i) because of its ongoing decline due to natural and anthropogenic factors and its reduction in range, and because the remaining population is small and fragmented, with an estimated current population of no more than 1048–2000 individuals [6,7,8]. In Chile and Argentina, the Patagonian huemul is classified as Endangered in the Red Data Books of Vertebrates, and it is also listed in Appendix I of CITES and UNEP/CMS conventions [7]. Some conservation measures currently prioritized by government agencies in both countries include increased efforts to obtain more information on Patagonian huemul subpopulations, such as their current distribution, abundance, and threats; encouraging more effective protection of the identified subpopulations, the creation of private protected areas with the presence of Patagonian huemul (or suitable habitats) to facilitate connectivity and dispersal, and the use of training to improve local skills in wildlife management and monitoring techniques [7,8]. Furthermore, a recovery strategy for Patagonian huemul must also include ex situ conservation initiatives, which provide valuable scientific data and animals for re-introduction programs [7]. There is currently only one private center for the conservation and reproduction of Patagonian huemul, located in the region of Los Ríos, in southern Chile (39°51′ S, 71°57′ W). This center was created to establish the first captive breeding project, with the main objective of reintroducing individuals in the future, and it is expected that other centers will be opened in the future [6]. To date, there have been no published protocols for assessing the welfare of this deer species, which would improve both the quality of life of captive individuals and conservation. Therefore, the aim of this study was to design and develop the basis of a welfare-assessment protocol (desktop study) for captive Patagonian huemuls in conservation centers.

## 2. Materials and Methods

The protocol is based on the WQ framework for cattle and was adapted for use in Patagonian huemuls because of its widely accepted structure, comprehensive design, and potential application to other species in managed care [3,9]. Farm-animal-welfare protocols were applied to ungulate species previously [3]. Of the 12 WQ criteria, three were modified considering current knowledge of the biology of the species, the difficulty of evaluation, and some new protocol proposals for other ungulate species [3,10].

The indicators in the assessment protocol were developed predominantly using peer-reviewed published research. To obtain information about wild and captive Patagonian huemul health, behavior, and ecology, as well as information on the welfare, health, and behavior of other deer species, we used the Google Scholar and Web of Science^TM^ search engines. The keywords *Hippocamelus bisulcus*/huemul/deer/ungulate were used in addition to the key concepts of each criterion. The terms hunger, thirst, minerals, “thermal comfort”, “ease of movement”, enclosure, disease, injury, “social behavior”, behaviors, “animal handling”, and “social environment” were used in the search. Keywords related to the species or its taxonomy were also merged with the following keywords related to potential indicators: “body condition”, “mineral supplementation”, shade, shelter, enclosure, quarantine, “nasal discharge”, “ocular discharge”, “hampered respiration”, diarrhea, lameness, “integument alterations”, “affiliative behaviour”, “intra-specific aggression”, stereotypies, “environmental enrichment”, “caretakers training program”, capture, immobilization, and handling. Furthermore, sources related to the topic found as references in other articles, or suggested by the search engines, were also included.

A total of 64 papers were reviewed. We found useful information to add to the protocol in seventeen of these papers, of which three provided useful detailed information on the behavior and general biology [6,11,12], and eight provided useful knowledge on the health of the species [13,14,15,16,17,18,19,20]. We also carried out searches in libraries of institutions in Chile and Argentina, where different documents provided information on historical conservation plans for this species. 

This protocol is mainly intended to be used to identify welfare problems in endangered species in the context of conservation centers. Thus, aggregation of different measures to obtain a global score is not proposed. However, a scoring system that assigns a value on a 0–2 scale (0 = no welfare concern, 1 = welfare concern, 2 = urgent welfare concern) is proposed for each category. This would facilitate the monitoring of welfare evolution over time and/or to compare different conservation centers in the future to identify the best care practices.

For this protocol, “animal group” should be understood as all the animals that are in the same facility, as a family group, or as unrelated individuals.

## 3. Results

### 3.1. Development of the Welfare Protocol

The protocol developed for the welfare assessment of captive Patagonian huemul included four principles, twelve criteria, and twenty-three indicators (Table 1). The twelve criteria of the WQ protocol were reduced to nine. One farm-related criterion was excluded (the absence of pain induced by management procedures), since handling and contact between humans and Patagonian huemul tend to be absent or very limited for reintroduction purposes. Moreover, the type of handling performed on these animals did not include any potentially painful practices. The criterion of comfort around rest was removed because of the difficulty of its measurement in field conditions. The criterion “positive emotional state” was also removed because there is a lack of valid indicators in this species. In addition, the criteria “presence of minerals in the diet” and “adequate social environment” were added, as they were considered essential for welfare assessment in captive Patagonian huemuls, as well as measurable. A description of the 12 criteria together with their proposed indicators and justifications is presented in the following sections.

#### 3.1.1. Absence of Prolonged Hunger

Both poor and excessive (excess of body fat) body conditions are indicative of a welfare problem and have been previously described as management concerns for captive deer [21]. To date, no body-condition scale has been developed for Patagonian huemuls. Therefore, the guidelines for the evaluation of this indicator in another deer (*Cervus elaphus*), which use a five-point scale [22,23], were followed, although they were modified and reduced to a three-point scale.

The adult animals (older than one year) were visually evaluated from the lateral and posterior perspectives. A scale of 0 to 2 was used. The values were defined as follows: 0 = adequate body condition, in which the pelvis, ribs, and spine were not easily distinguished or appeared rounded rather than sharp, and the rump area was flat or slightly convex; 1 = leanness, in which the pelvis, ribs, and spine were prominent and the rump was concave; 2 = animal with cachexia (croup very concave, column visible) or in a fat condition (the wings of the pelvis were concealed under a thick layer of fat; sacral spinous processes were well enveloped, and the rump areas were convex).

#### 3.1.2. Absence of Prolonged Thirst

Ad libitum access to high-quality water is one of the most important welfare requirements for most animals [10]. Ideally, Patagonian huemuls must have access to watercourses, as in the wild [24,25]. The presence and availability of water sources throughout the year for each animal group should be visually evaluated. The indicator was scored as either 0 (the presence of at least one permanent water source per animal group) or 2 (the absence of at least one permanent water source per animal group).

#### 3.1.3. Presence of Minerals in the Diet

To maintain normal physiological functions, wild ungulates often seek sources of salt [10]. Due to the habitat conditions of the Patagonian huemul, some studies describe micronutrient deficiencies (essential minerals e.g., Se, Cu, Mn, and others) in this species, which affect their growth, skeletal development, reproduction, and immuno-competence [26,27,28]. In addition, it has been observed that Patagonian huemuls that can access blocks of mineral salts have better coat and antler conditions, and females give birth to heavier offspring [29]. The current protocol included one indicator (3.1) that was assessed for each animal group. At least two blocks of mineral salt (3 kg) were visually inspected. The scoring system was 0 when mineral salt blocks were present, and 2 when no blocks were present.

#### 3.1.4. Thermal Comfort

In captivity, there may be long-term exposure to temperatures that are different from those in the natural habitat, which can eventually result in thermal stress in animals [3]. Due to its natural habitat, the Patagonian huemul is better adapted to tolerate low-to-medium temperatures than high temperatures. In general, wild and captive deer require protection from direct sunlight [30,31]. In winter, the Patagonian huemul seeks refuge in mixed forest and grassland areas [6].

Two indicators were developed to assess thermal comfort: the availability of shade (4.1) and shelter (4.2). To assess these indicators, we recorded whether all the animals in each facility had access to adequate shade and shelter simultaneously. The presence of natural shadows (e.g., trees), scrub areas, forests, and artificial structures in conservation centers were visually evaluated. Satellite-type botanical records and geographical charts are required to determine the availability of resources throughout the year. The enclosure was scored as follows: 0 “presence of resources, simultaneous access to adequate shade and shelter”; 1 “presence of resources, which do not, however, meet best practice, as described in category 0”; and 2 “absence of resources”.

#### 3.1.5. Ease of Movement

Limited information is available on the minimum space per animal in captivity for Patagonian huemuls [24]. Recommendations suggest that areas of up to 50 ha are needed for semi-captivity, but densities or other parameters in relation to the number of individuals have not been published [32]. Other studies on national parks in Chile reported 309 ha or 3 km^2^ per established family group of up to four individuals [12]. The only Patagonian-huemul-conservation center in Chile (the Huilo Huilo Reserve) has sixty-four hectares for seven adult animals (density 10/km^2^), with good results in terms of breeding success [25]. Although we are fully aware that reproductive success alone should not be considered a guarantee of good welfare [1], we took these data as a starting point.

An indicator was developed to assess ease of movement for each animal group, as follows: 0 “size of the enclosure > 64 hectares”; 1 “size of the enclosure > 50 and < 64 hectares”; 2 “size of the enclosure < 50 hectares” per animal group.

#### 3.1.6. Adequate Enclosure Standards

The perimeter fence of a Patagonian huemul enclosure must be at least 3.5 m high, it must have double wiring with material approved for deer, with electrified wire in the upper part, single in the middle, and double in the lower part, and it must operate permanently [25]. It must have vegetation inside and no trees outside, which would allow predators to enter [32]. The availability of quarantine facilities must be ensured to enable effective captive management and breeding programs [10]. The quarantine for all species must be supervised by a veterinarian and must consist of a minimum of 30 days, unless otherwise advised by the appointed veterinarian [33].

Two indicators were developed. The presence of a perimeter fence and an area exclusively dedicated to quarantine should be visually evaluated, as should conditions in terms of measurements and materials. Satellite-type photographic and botanical records are required to determine the daily availability of vegetation in the internal part of the perimeter of the entire fence throughout the year. In addition, annual records of animal arrival, the time spent in the area, and the veterinary medical professionals responsible for inspection and handling should be requested. The enclosure is scored as follows: 0 “presence of perimeter fence and quarantine area”; 1 “presence of resources, which do not, however, meet best practice regarding measurements, materiality and/or supervision”; and 2 “absence of resources”.

#### 3.1.7. Absence of Disease

The WQ protocols for ruminants include indicators that can be used to assess gastrointestinal and respiratory conditions through remote observation [3]. There are many reports of findings of different pathogens in populations of Patagonian huemul, such as viral diseases like bovine viral diarrhea [13] and parasitism [14,15], which mainly due to contact with domestic animals and the introduction of exotic herbivores. Patagonian huemul feces are small, rounded, and black in color, like those of sheep; however, depending on the animals’ diet, they can be cylindrical in shape [34]. In wild deer, including the Patagonian huemul, the presence of Mycobacterium avium subsp. paratuberculosis can cause the disease paratuberculosis which can present—among other clinical signs—with diarrhea [18,19].

Four indicators were proposed: “nasal discharge” (7.1), “ocular discharge” (7.2), “hampered respiration” (7.3), and “diarrhea” (7.4). The indicator “ocular discharge” was evaluated on both eyes, as described for horses [35]. The indicator “diarrhea” was developed as described for sheep and goats [36,37]. The other health indicators were developed following the recommendations given for cattle (4). If any indicators were observed, the animals were assessed as 2 “presence of the sign” and 0 “no sign of these indicators”.

#### 3.1.8. Absence of Injuries

Two indicators were developed: lameness (8.1) and integument alterations (8.2). Hoof problems are common in ruminants [3]. Some causes of lameness have been described in Patagonian huemuls, which show clinical signs of variable severity [20]. The moving animals were visually observed from behind and from the side on a flat, non-slippery surface. The rhythm of the steps was evaluated, as well as the weight borne by each limb [4]. The animals were scored as 0 = “no lameness, normal rhythm, and support; 1 = “moderate lameness, unnormal rhythm, only one leg affected”; and 2 = “severe lameness, alteration when bearing weight on one limb, or more than one leg affected”.

Integument alterations, such as patches and lesions without hair or swelling, can be consequences of disease, rough handling, intraspecific aggression, interspecific aggression, or an inappropriate physical environment [3]. However, caution must be exercised, since the Patagonian huemul is a species that presents molting, or changes to the coat, twice per year, in autumn and spring [6,34]. Recently, infectious and parasitic diseases were shown to cause alterations in integuments in different regions of the body [6,16,17].

Only skin changes with a minimum diameter of 2 cm were counted. Hairless patches included areas of hair loss with undamaged skin and extensive thinning of the coat owing to parasites and hyperkeratosis. Injury or swelling included skin damage in the form of scabs or wounds, dermatitis due to ectoparasites, and ear injuries due to torn ear tags. Without touching the animals, three body regions on both sides of the animal were examined: the body, hind leg, and front leg. These body regions were scanned from back to front, excluding the lower side of the abdomen and the inner side of the legs, but including the inner side of the opposite hind leg. The animals were scored as follows: 0 = “no alteration of the integument (no hairless area, no injury, no inflammation)”; 1 = “minor integument alterations: at least one patch without hair, but without injury/swelling”; and 2 = “severe integument alterations: at least one lesion/swelling or a large hairless patch”.

#### 3.1.9. Expression of Social Behaviors

Patagonian huemuls form small groups of two to four individuals, which can be permanent or transitory, with little or no sexual segregation. The minor difference in bodyweight between males and females is atypical in medium-sized deer and could explain the difference in social behavior between this and other deer species [6]. In the wild, different affiliative behaviors have been described, including those between males and females in the reproductive stage, between mothers and their offspring, and between juvenile individuals [12].

Antagonistic behaviors must be recorded to assess negative social interactions between the members of a group. These behaviors include physical interactions, vocal communication, fighting, and/or chasing [12]. In family groups of Patagonian huemuls in the wild, expulsion behavior has been observed, which consists of an adult chasing a young animal (older than one year of age) within the territory to initiate its independence [12].

Two indicators were developed: “affiliative behavior” (9.1) and “intraspecific aggression” (9.2). An ethogram (Table 2) was developed based on the social behavior of Patagonian huemul [6,11,12], and information on the behavior patterns described for other species was also used [3,4,10]. The overall observation time was 180 min per group in continuous 20 min sessions [3].

Depending on the time of year when the evaluation was conducted, the following classifications (Table 3) were created to evaluate the interactions between the animals.

#### 3.1.10. Expression of Other Behaviors

A behavior that is invariable, repetitive, and lacking in clear purpose is called a stereotyped behavior [3]. To date, no precedent for stereotypical behaviors have been observed in captive Patagonian huemuls. However, wild ungulates in captivity have been reported to be prone to developing oral stereotypies [1,38]. In captive ungulates, object-licking, dirt-eating, tongue-rolling [39] fence-biting [10] have been observed. This protocol proposes that all animals should be assessed for the presence or absence of “stereotypical behaviors” (10.1) during the observation of the expression of social behavior. The animals were scored as 0, for “stereotypes absent,” and 1, for “stereotypes present”.

An enriched environment must be provided to all animals under human care, including those that are reintroduced into the wild [10]. Patagonian huemuls prefer to live in mixed areas in the presence of forests and open spaces, with shrubs and grasses [6]. Although there are few management guidelines for this species [24], guidelines for other ungulate species recommend providing opportunities for animals to perform natural browsing, grazing, fleeing, and/or hiding behaviors. To measure environmental enrichment (10.2), the presence of structural components, such as rocks and irregular terrain, vegetation, forests, and grasslands should be evaluated, and the use of these resources must be recorded through physical presence, grazing, and browsing. The enclosure was scored as follows: 0 = “the presence and use of resources”; 1 = “the presence of resources, whose use is not observed during the evaluation”; and 2 = “lack of resources”.

#### 3.1.11. Good Animal Handling

Two indicators were developed to assess this criterion: the caretaker-training program (11.1) and capture, immobilization, transport, and handling (11.2). There should be a technical and veterinary team with experience and resources to check and, if necessary, treat sick animals. The avoidance of stress is the main consideration and, thus, human contact with animals is kept to a minimum [40]. The training of personnel who monitor and manage Patagonian huemuls in a semi-confinement center is of the utmost importance to meet the objectives set for the reproduction and subsequent reintroduction of this species [25] and for protecting individual welfare. 

The presence of evidence of training and education programs and protocols should be evaluated. Knowledge of the natural history of this species, the basis of animal welfare, monitoring strategies, and the management of animals in semi-confinement, as well as the medicine used to treat wild ungulates, must be accredited. The enclosure was scored as follows: 0 = “evidence of knowledge of the species, of monitoring strategies and the management of animals in semi-confinement, and of medicine for wild ungulates”; 2 = “the absence of evidence of knowledge about one or several areas of training”.

The methods used to capture, immobilize, transport, and manipulate animals must be reviewed and approved by the animal health authority of the country where the conservation center is located before performing these procedures [40,41]. In addition, the characteristics and age (juveniles or adults) of Patagonian huemuls must be considered, and a record of all the actions that are undertaken must be kept [12,42]. In addition, the presence of updated best-practice protocols for capture, immobilization, and handling must be recorded. These protocols should be refined to reduce stress, fear, health effects, and other negative effects of these activities on animal welfare. The veterinarian responsible for the center must sign the protocol. The enclosure was scored as follows: 0 = “the presence of protocols and registers”; 2 = “the absence of adequate protocols and/or records”.

#### 3.1.12. Adequate Social Environment

Three indicators were developed: the number of Patagonian huemuls, the composition of the group, and the presence of animals of other species. Patagonian huemuls are gregarious deer characterized by the formation of small groups, which can be permanent or transitory [12]. Permanent family groups are commonly made up of two to four members [6], and transitory groups of one to eleven individuals can be found, depending on the season of the year [43,44]. The number of Patagonian huemuls (12.1) per group was recorded. The enclosure was scored as follows: 0 = “the presence of records of the number of individuals per established animal group and an adequate number of individuals per family group (a minimum of two animals)”; 2 = “no registration of the number of individuals per animal group and/or an inappropriate number of individuals per family group (a solitary individual for more than 3 months)”.

Group size is not the only factor to consider, as group composition and compatibility between animals are also important [3]. A permanent family group of Patagonian huemuls can include at least one male, one female, and one or two offspring of different ages [12]. The composition of the animal groups (12.2) in the center should be evaluated visually and based on records, and the sex and age of every animal must be recorded. The enclosure was scored as follows: 0 = “the registration of an adequate composition in the group(s) at the center (one male and one female, with or without offspring)”; 2 = “the absence of records or inadequate group composition”.

The presence of predatory animals, such as pumas (*Puma concolor*), foxes (*Lycalopex culpaeus*/*Lycalopex griseus*), and feral dogs (*Canis familiaris*) can negatively affect the welfare and group dynamics of Patagonian huemuls [25,42]. In addition, the presence of cattle [45] and other species of introduced deer (*Cervus elaphus*), can be a risk factor due to competition for food resources and disease transmission [46]. When assessing the presence of animals of other species (12.3), conservation centers must ensure that their perimeter fences can prevent the entry of other animals (cows, other deer, and predators). The enclosure was scored as follows: 0 = “the absence of other species of deer, no cattle, and no predators”; 2 = “evidence of the temporary or permanent presence of deer, cattle, or predators”.

## 4. Discussion

Traditionally, interest in wild animals has been directed primarily toward conservation [1]. However, in recent years, animal welfare has become an important aspect of captive-wild-animal management and ex situ conservation programs, not only for ethical and for legal reasons, but also to maintain healthy individuals and populations and ensure the success of programs, since many welfare problems have a negative effect on reproduction [1,2]. Conservation approaches based solely on the protection of some sites and observational studies are insufficient to prevent the Patagonian huemul from becoming extinct. Current information gaps can be bridged by investigating Patagonian huemuls in semi-captivity [7,8,47].

Since many factors can affect welfare, a holistic approach with multiple indicators is necessary to obtain a clear picture of an animal’s welfare status [3,48,49]. This welfare protocol for Patagonian huemuls is the first documented work toward the development of a standardized welfare-assessment tool for this species, which is in danger of extinction [8]. Other protocols have been developed specifically for other wild species in captivity, based on the WQ framework [3,50]. In the case of Dorcas’s gazelles, the protocol included 10 criteria and 23 indicators, and the protocols for foxes (*Vulpes* spp.) and mink (*Neovison vison*) both included 12 criteria, but with 26 and 22 indicators, respectively. Thus, the proposed protocol for the Patagonian huemul with 12 criteria and 23 indicators does not seem to differ significantly from other WQ-based protocols created for use in other species.

In protocols used to assess animal welfare, the criteria and indicators do not always have the same ease of application. Although it is an important criterion for inclusion in the welfare-assessment protocol, positive emotional state was not included. This was due to the lack of current research in conservation centers or zoos concerning captive ungulates and, thus validated, of indicators [3,10]. Moreover, and for the same reasons, many indicators of positive welfare were not included. However, indicators to assess the emotional state and/or positive welfare of Patagonian huemuls should be added in the future, as when knowledge of the emotional states and/or positive welfare of ungulates increases, validated indicators will appear. 

Another criterion that was not included in this protocol was comfort while resting. Several behavioral indicators are used to evaluate comfort around resting, including rising and lying-down movements (such as the time spent lying down, the frequency of lying bouts, and the duration of individual bouts) [4,49]. However, to evaluate these indicators, it is necessary to observe animals when they perform resting behaviors, which is difficult with Patagonian huemuls due to the presence of vegetation. However, it was found that, when lying down, Patagonian huemuls first drop one of their front knees, and then the other, before they reach a “kneeling” position [34], and that, during rest, the Patagonian huemul observes, sleeps, and “ruminates” [44]. In addition, a seasonal resting pattern has been described for the Patagonian huemul in the wild [44], with variable rest throughout the seasons of the year, with three periods in spring–summer (for a total four to seven hours), while in autumn–winter, this is reduced to two shorter periods (between 4 and 5 3/4 h), which has not been described in captive conditions. Therefore, observations are required to create a new comfort-around-resting indicator in the future.

The goal of protocols aimed at assessing the welfare of wild animals kept in captivity is their regular use as management tools in the centers where these animals are held [3]. The protocols developed using the WQ protocol for farm animals as a reference for different species differ in terms of the time required for their implementation [10]. The protocol developed for minks and foxes requires three visits to each farm [50]; for bottlenose dolphins, the protocol requires two days for the complete welfare assessment of each dolphin pod, which include up to 10 individuals [9]. In the case of protocols for Dorcas’s gazelles, which require less than six hours per herd (17 individuals) [3], while the protocols for the Punjab urial require 5 h for the complete assessment of 23 individuals [10]. The proposed welfare protocol was designed to practical and was executed two days for every four Patagonian huemuls. This is because the smallest period to evaluate social behavior (affiliative and aggressive) was 180 min, in continuous sessions of at least 20 min per group of animals. During this time, it was also possible to evaluate the expression of other behaviors, such as stereotypies and the presence and use of resources in an enriched environment. In addition, we must consider in this protocol that some indicators may be difficult to evaluate with precise results, and that the evaluation times for these indicators may be longer, especially in animals of wild origin such as the Patagonian huemul, when they are kept in enclosures with dense vegetation. For example, measurements of body conditions and health indicators can take longer than in other species, given the difficulty in adequately observing each animal in a wide area, which may be covered with vegetation.

Measurements can be broadly separated into those carried out by visual observation while animals go about their daily activities, those assessed partly by consulting the person in charge of the conservation center and its records, and those performed opportunistically. Systematic behavioral monitoring is essential to achieve the highest standards of welfare, and, therefore, behavioral assessments of Patagonian huemuls can be applied regularly as management tools, with or without a full assessment, as described for other types of animal [9]. In principle, animal-welfare assessments should be conducted by trained persons familiar with the relevant methodology, metrics, and evaluation tools [9]. Therefore, those responsible for conservation centers should be trained to use these tools effectively.

This protocol is simple, practical, and systematic. However, practical applications are conducted in the field, they must be accompanied by a complete instruction manual with information, such as technical sheets and photographic and video references. Although the Patagonian huemul is the only South American deer in danger of extinction according to the IUCN [8], it is the species with the lowest amount of known information [47]. Our protocol is presented as an initial step in assessing the welfare of Patagonian huemuls in captivity; however, its validation has not yet been completed. Among the challenges for the application of this protocol is the need to evaluate each indicator under field conditions, establish and/or adjust evaluation times, and use powerful equipment, such as the latest generation of binoculars and high-resolution cameras. Despite its need for validation, the protocol in this study could easily be adapted to other deer species in conservation centers, which would hopefully lead to the development of new welfare-assessment protocols, thus expanding the currently small field of deer welfare.

## 5. Conclusions

We developed a protocol for the assessment of the welfare of Patagonian huemuls in conservation centers, using a protocol for the assessment of welfare in cattle as a base, and applying modifications and species-specific adaptations. The first specific protocol developed for Patagonian huemuls comprised four basic principles, twelve criteria, and twenty-three animal- and resource-based indicators. This protocol requires validation in conservation centers (reproduction, rescue, rehabilitation, or treatment centers), which may be established in the future to preserve this species. In addition, this protocol can be used as a basis for developing welfare protocols for other deer species.

## Figures and Tables

**Table 1 animals-13-02495-t001:** Principles, criteria, and indicators of the protocol to assess welfare in captive Patagonian huemuls.

Principles	Criteria	Indicators	Indicator Type
Good Feeding	1. Absence of prolonged hunger	1.1 Body condition	Animal-based
	2. Absence of prolonged thirst	2.1 Availability of water	Resource-based
	3. Presence of minerals in the diet	3.1 Adequate mineral supplementation	Resource-based
Good Housing	4. Thermal comfort	4.1 Availability of shade 4.2 Availability of shelter	Resource-based Resource-based
	5. Ease of movement	5.1 Enclosure size	Resource-based
	6. Adequate enclosure standards	6.1 Perimeter fence 6.2 Quarantine zone	Resource-based Resource-based
Good Health	7. Absence of disease	7.1 Nasal discharge 7.2 Ocular discharge 7.3 Hampered respiration 7.4 Diarrhea	Animal-based Animal-based Animal-based Animal-based
	8. Absence of injuries	8.1 Lameness 8.2 Integument alterations	Animal-based Animal-based
Appropriate Behavior	9. Expression of social behaviors	9.1 Affiliative behavior 9.2 Intra-specific aggression	Animal-based Animal-based
	10. Expression of other behaviors	10.1 Stereotypies 10.2 Environmental enrichment	Animal-based Resource-based
	11. Good animal handling	11.1 Caretaker-training program 11.2 Capture, immobilization, and handling	Resource-based Resource-based
	12. Adequate social environment	12.1 Number of Patagonian huemuls 12.2 Composition of the group 12.3 Presence of animals (other species)	Resource-based Resource-based Resource-based

**Table 2 animals-13-02495-t002:** Description of the social behaviors (affiliative and aggressive) that are included in the welfare protocol for Patagonian huemul (Adapted from [3,4,6,10,11,12]).

Type of Behavior	Behavior Pattern	Description of Behavior
Affiliative behavior	Mutual grooming	The animal brushes with its muzzle any part of the body of another member of the group, except for the anal region or the prepuce. If the animal stops brushing the receiver for more than 10 s and then starts brushing the same receiver again, this is recorded as a new bout. A new bout is also considered if the actor starts brushing another receiver, or if there is a role reversal between the actor and the receiver.
	Social smelling	The animal smells any part of the body of another member of the group, except for the anal region or the prepuce. If the animal stops smelling for more than 10 s and then starts smelling the same receiver again, this is recorded as a new bout. A new bout is also considered if the actor starts smelling another receiver, or if there is a role reversal between the actor and the receiver.
	Licking	One animal licks any part of another animal with the tongue, except for the anal region or urine. If the actor animal stops licking for 10 s and starts again, this is to be counted as a new bout, regardless of whether the actor licks the same receiver or another. If the actor receives brushing from the receiver, this should also be counted as a new bout.
	Suckling	The behavior of calves while consuming milk from the udder. A phase during which a calf is allowed to suckle milk from the dam.
	Horning	The animals rub foreheads, horn bases, or horns against each other’s head or neck without obvious harmful intention. Neither of the opponents takes advantage of the situation to become victorious. A new bout is considered if the same animals start horning after 10 s or more, or if the horning partner changes.
Aggressive behavior	Displacement with physical contact	The actor buts, hits, thrusts, strikes, pushes, or penetrates the receiver with the forehead, horns, horn base, or any other part of the body in a forceful movement, resulting in the receiver giving up its position.
	Displacement without physical contact	The actor threatens or interacts with the receiver without making physical contact, resulting in the receiver giving up its position.
	Chasing	The actor makes an animal flee or give up its current position by following it rapidly or running behind it, sometimes with added threats, like jerky head movements. Chasing is recorded even if it is not followed by an interaction with physical contact.
	Fighting	Two contestants vigorously push their heads (foreheads, horn bases, and/or horns) against each other while planting their feet on the ground, both exerting force against each other. A new bout starts if the same animals restart fighting after more than 10 s, or if the fighting partner changes.

**Table 3 animals-13-02495-t003:** Description of the social interactions between Patagonian huemuls (adapted from [6,11,12]).

Interaction	Scale = 0	Scale = 1
Adult male-female interactions	Male and female rest together. There is vocal communication. Mating behaviors (flehmen, smelling, licking, and mounting) are seen in the reproductive season.	Absence of affiliative behavior and/or mating behaviors in the reproductive season.
Fawn interactions	Female maintains recurrent contact with the fawn by smelling, touching, and licking. This can also be seen in vocal communication and/or playing. If the juvenile is still in the group with the fawn, occasionally play is also observed.	Absence of mother–fawn and/or fawn–yearling (juvenile) affiliative behavior.
Interaction between juvenile or adult individuals in the group	There is licking and/or smelling, and/or “horning” behavior.	Absence of affiliative behavior.
Adult male–male interactions	Absence of antagonistic behavior: snorting, stomping, trashing bushes and/or fighting.	Presence of antagonistic behavior, including snorting, stomping, trashing bushes, and/or fighting.
Yearling interactions	Absence of expulsion behavior (adult chasing a yearling).	Presence of expulsion behavior (adult chasing a yearling).

## Data Availability

Not applicable.

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
