# Peer review of "A Protocol to Assess the Welfare of Patagonian Huemul (Hippocamelus bisulcus) in Conservation Centers"

_animals, 2023, doi:10.3390/ani13152495_

Round 1
Reviewer 1 Report
"A Welfare Protocol to Assess the Welfare of Huemul (Hippocamelus bisulcus) in Conservation Centers" Under the background of the conservation center, based on the welfare quality (WQ) framework of cattle, a baseline welfare plan was made for the welfare evaluation of Huemul in the conservation center, including 4 principles, 12 standards and 23 indicators based on animals and resources, and the welfare problems of endangered species were studied. The index system of welfare evaluation is feasible, which is of great significance to the protection of this species and a beneficial supplement to the construction of the standard system of wildlife welfare evaluation. The overall thinking of the paper is clear, the conception is correct, the scientific problems are clear, the writing logic of the paper is rigorous and the writing is standardized. But there are the following shortcomings:
1. In line 13, the first letter of "(Hippocamelus bisulcus)" should be capitalized.
2. It is suggested that "deer" be deleted in Keywords.
3. The background is unclear, so it is suggested to add "the research status of animal welfare assessment methods at home and abroad" and "Current status of huemul".
4. There is lack innovation in introducing the background of this study.
5. In terms of research method, it is too small to search two literature retrieval databases closely, and it is suggested to expand the search scope.
6.3.1 Are there any corresponding references for the three newly added standards such as "presence of minerals in the diet" and "Adequate social environment"?
7. Table 1 and Table 2 recommend adding a column of references corresponding to each indicator
8. It is suggested to distinguish minerals in 3.1.3, because the human body can be divided into two categories: essential and non-essential. What kind of minerals are you referring to here?
9. In the second paragraph of the discussion, it is suggested that the research status of other captive wild species evaluation based on WQ should be increased and compared with this research index.
10. There are grammatical errors in the text. It is suggested to modify and polish the whole text.
11. The references should be unified.
There are grammatical errors in the article
Author Response
Thank for your comments and suggestions. Please see the attachment.
Best regards,
Enrique, Xavier, Oriol.

Reviewer 2 Report
The manuscript reports an interesting desktop study that develops a welfare assessment tool to guide a likely future captive breeding programme for the endangered Huemul. The assessment criteria are logical and well-researched from a limited literature on this species, but the results remain untested. Thus, it is questionable as to whether the study is more than an addition to a conservation plan for the huemul. In favour of publication is the methodology that could guide other welfare assessments of this kind, and the adaptation of the WQ protocol for a non-domesticated ungulate that in the first generation will be wild caught. Thus, on balance I favour publication.
The manuscript is very well-written but I suggest a few corrections as follows:
Line 107: It is unclear what “excessive body conditions” means?
Line 143: In winter, the huemul seeks
Line 173: ensured to enable effective
Line 181-182: It unclear what “animal income” means?
Line 189: and parasitism
Line 200: individuals and so in
Line 252 entry for Horning: The head is the point of physical contact
Line 252 entry for Fighting: 10 s (consistency in table)
Well-written with some minor errors and oddities.
Author Response
Thank you for your comments and suggestions,
Best regards,
Enrique, Xavier, Oriol.

Reviewer 3 Report
Perhaps not necessary to use welfare twice in title and in Abstract:
A Welfare Protocol to Assess the Welfare of Huemul (Hippocamelus bisulcus) in Conservation Centers
The topic of the paper will no doubt find readership within and outside the 'Animals' community. The paper, however, will better serve its audience with a robust revision to strengthen the clarity of the writing the content of the study itself.
A few key points to focus on for improvement include:
1. Better organizing the content of the manuscript. For example, there is information in the Discussion that is better suited to the Introduction. In fact much of the information in the opening paragraphs of the Discussion is more appropriate for the Introduction. The Discussion shouldn’t really be “introducing” new concepts.
2. On this note, definitions of welfare should come in the Introduction or the Methods at the very latest (not in the Discussion).
3. Welfare and wellbeing are used somewhat interchangeably. Conceptually and operationally these are not the same. The authors should clarify this
4. The Methods section is too vague/imprecise.
5. It should be clear from the outset that the protocol was developed as a desktop study and not from an empirical one. This should be made clear in the title and Introduction. Thus, what the authors mean by protocol must be made clear.
6. There are some misleading portrayals of the history of welfare concerns. Historically, welfare of captive animals (including wild animals) precedes formal/systematic protocols for assessing free-living wild animals. See opening paragraph of Discussion.
7. Authors should consider positive aspects of welfare and wellbeing in any assessments – not just in avoiding negative states. What would the scoring look like in this case?
See my other comments
Author Response
Thank you for your comments and suggestions,
Please see the attachment.
Best regards,
Enrique, Xavier, Oriol.

Reviewer 4 Report
Please see file attached.

Author Response

(The authors gave the same response as above.)
